# Clinical Significance and Regulation of ERK5 Expression and Function in Cancer

**DOI:** 10.3390/cancers14020348

**Published:** 2022-01-11

**Authors:** Matilde Monti, Jacopo Celli, Francesco Missale, Francesca Cersosimo, Mariapia Russo, Elisa Belloni, Anna Di Matteo, Silvia Lonardi, William Vermi, Claudia Ghigna, Emanuele Giurisato

**Affiliations:** 1Department of Molecular and Translational Medicine, University of Brescia, 25100 Brescia, Italy; matilde.monti@unibs.it (M.M.); f.missale@unibs.it (F.M.); silvia.lona@gmail.com (S.L.); william.vermi@unibs.it (W.V.); 2Department of Biotechnology Chemistry and Pharmacy, University of Siena, 53100 Siena, Italy; jacopo.celli@student.unisi.it (J.C.); francesca.cersosi@student.unisi.it (F.C.); mariapia.russo@student.unisi.it (M.R.); 3Department of Head & Neck Oncology & Surgery Otorhinolaryngology, Antoni Van Leeuwenhoek, Nederlands Kanker Instituut, 1066 Amsterdam, The Netherlands; 4Istituto di Genetica Molecolare “Luigi Luca Cavalli-Sforza”, Consiglio Nazionale delle Ricerche, 27100 Pavia, Italy; elisa.belloni@igm.cnr.it (E.B.); anna.dimatteo@igm.cnr.it (A.D.M.); 5Department of Pathology and Immunology, Washington University School of Medicine, St. Louis, MO 63130, USA; 6Division of Cancer Sciences, School of Medical Sciences, Faculty of Biology, Medicine and Health, The University of Manchester, Manchester M13 9PL, UK

**Keywords:** ERK5, cancer, alternative splicing, prognostic markers, tumor microenvironment, cancer-stem cells

## Abstract

**Simple Summary:**

Alterations of the MEK5/ERK5 signaling pathway have been described in several tumors, and emerging data are trying to explain the complexity of ERK5-related mechanisms in both physiological and pathological conditions. Here, we provide an overview of the known evidence about ERK5 expression, activity, and regulation in tumors. Moreover, our study investigated the clinical significance and the prognostic value of ERK5 deregulation in human cancer. Upregulation and overexpression of ERK5 were reported in several tumor types associated with advanced stage, metastases, and worse overall survival. In addition, we focus on emerging insights regarding the relevance of ERK5 in a tumor-associated microenvironment. Finally, the recently discovered ERK5 isoforms, correlated with pro-tumoral function and negative regulation of ERK5, were also described.

**Abstract:**

Extracellular signal-regulated kinase 5 (ERK5) is a unique kinase among MAPKs family members, given its large structure characterized by the presence of a unique C-terminal domain. Despite increasing data demonstrating the relevance of the ERK5 pathway in the growth, survival, and differentiation of normal cells, ERK5 has recently attracted the attention of several research groups given its relevance in inflammatory disorders and cancer. Accumulating evidence reported its role in tumor initiation and progression. In this review, we explore the gene expression profile of ERK5 among cancers correlated with its clinical impact, as well as the prognostic value of ERK5 and pERK5 expression levels in tumors. We also summarize the importance of ERK5 in the maintenance of a cancer stem-like phenotype and explore the major known contributions of ERK5 in the tumor-associated microenvironment. Moreover, although several questions are still open concerning ERK5 molecular regulation, different ERK5 isoforms derived from the alternative splicing process are also described, highlighting the potential clinical relevance of targeting ERK5 pathways.

## 1. Introduction

In the MAPK signaling network, ERK5 contains many features that are structurally and functionally distinct from other MAPKs. Although its N-terminal kinase domain has high homology with ERK1/2, ERK5 has a unique and extended C-terminus containing a transcriptional activation domain (TAD), enabling active ERK5 to undergo autophosphorylation of its TAD, thereby exerting direct control over gene transcription [1]. Activation of the ERK5 signaling cascade has been emerging as an important mediator of cell proliferation through induction of several cell cycle molecules, including c-MYC, cyclin D1, n-MYC, SGK, RSK2, and NF-κB [2,3,4,5,6,7], to regulate the expression of c-JUN, a proto-oncogene required to cell growth [8,9] and the activation of survival signaling [10]. It has also been demonstrated that ERK5 can downregulate CDK-inhibitors’ expression, thus enabling cells to overcome the G1-S phase [11,12]. Because of that, various studies have addressed the involvement of the MEK5/ERK5 route in cancer, having found that such route is deregulated in several neoplasias [12,13,14,15,16]. Pharmacologic tools and genetic approaches clearly demonstrated that targeting the MEK5/ERK5 route may be used therapeutically for cancer treatment.

Here, we provide an updated review of the existing evidence regarding the relevance of upregulated ERK5 expression and activity in promoting tumor growth and cancer stemness. We analyzed the clinical significance of ERK5 gene expression in cancers correlated with aggressive tumor states and poor patient outcomes. We then focused on emerging insights regarding the role of ERK5 in the tumor-associated microenvironment. In addition, reported evidence suggested the importance of ERK5 alternative splicing generating protein variants that could be involved in tumor growth.

### 1.1. ERK5 Structure

ERK5 is encoded by the Mitogen-Activated Protein Kinase 7 (*MAPK7*) gene (Figure 1).

The elevated molecular weight of ERK5 (110 kDa), which is almost double compared to other family members, also explains the name of Big Mitogen-Activated Protein Kinase 1 (BMK1). Structurally, ERK5 is constituted by an N-terminal half (1–406 aa) and a unique C-terminal tail (410–816 aa), which exerts an autoinhibitory function [17]. The N-terminus includes a region required for cytoplasmic targeting (1–77 aa) and a kinase domain (78–406 aa), which shares 66% of sequence identity to the one of ERK2 [18]. Of note, a Mitogen-Activated Protein Kinase 5 (MEK5) interacting sequence (78–139 aa) and an oligomerization domain (140–406 aa) have been identified within the kinase domain of ERK5 [19]. The C-terminus is constituted by two proline-rich (PR) domains, termed PR1 (434–465 aa) and PR2 (578–701 aa), which are considered to be potential binding sites for Src-homology 3 (SH3)-domain-containing proteins [18,19], a MEF-2-interacting region (440–501aa) [18], a nuclear localization signal (NLS) (505–539 aa), and a transcriptional activation domain (TAD) (664–789 aa) [20]. The latter is associated with the activation of several transcription factors, an ability unique to ERK5 with respect to the other MAPKs [21].

### 1.2. Regulation of ERK5 Activation

The MEK5-ERK5 signaling axis is triggered by several growth factors such as Vascular-Endothelial Growth Factor (VEGF), Epidermal Growth Factor (EGF), Fibroblast Growth Factor-2 (FGF-2), Platelet-Derived Growth Factor (PDGF) [22,23,24], and inflammatory cytokines like Interleukin-6 (IL-6) and Interleukin-8 (IL-8) [25], as well as osmotic and oxidative stress [26], which, in particular, can activate the MEK/ERK Kinases 2/3 (MEKK2/3) determining the phosphorylation of MEK5 within the activation motif (Ser311-Thr315). The activated MEK5 eventually mediates the phosphorylation of the TEY (Thr218-Glu-Tyr220) motif and determines the terminal activation of ERK5 [27]. Notably, the presence of PB1 domains organizes MEKK3, MEK5, and ERK5 into one signaling competent complex, via PB1 domain-mediated heterodimerization [28,29].

Moreover, RAS, the Proto-oncogene tyrosine-protein kinase Src (SRC), TPL/COT, and Protein kinase B (Akt) have all been reported to be MEKK2/3 and/or MEK5 upstream activators [30,31]. Particularly, it is known that the PhosphatidylInositol 3-Kinase (PI3K/Akt) pathway can trigger the activation of MEK5-ERK5 in neuroblastoma and malignant mesothelioma [7,32]. Furthermore, the Signal Transducer and Activator Of Transcription 3 (STAT3) upregulation has been reported to significantly increase the transcription of MEK5 in breast cancer (BC) [33]. It has been also demonstrated that TGF-β1 can mediate the association between MEK5, ERK5, and their downstream target MEF2C, via Anaplastic Lymphoma Kinase (ALK) receptor and the p38 mitogen-activated protein kinase, in HKC-8 transformed cells and human primary renal proximal tubule epithelial cells (PTECs) [34].

The overexpression of MEKK2/3 has been observed in different tumors such as colorectal, prostate, esophageal, breast, cervical, kidney, and lung cancer (LC) [35,36,37,38]; the latter one is also characterized by an increased MEK5/ERK5 signaling axis phosphorylation, and it has been recently demonstrated that pharmacological and genetic inhibition of both proteins, reduces LC cell proliferation [39].

MEKK2 and MEKK3 can activate distinct MAPK signaling pathways, but the specificity of MEK5 is restricted uniquely to ERK5 [18]. Opposed to the positive regulation of these protein kinases, protein kinase C (PKC) has been reported to inhibit ERK5 activation [40]. Furthermore, different phosphatases, such as the Phosphotyrosine-specific phosphatase (PTP-SL) and Dual-specificity phosphatases (DUSPs) family members, are involved in the downregulation of the ERK5 signaling axis. As a matter of fact, PTP-SL is thought to dephosphorylate the TEY microdomain, hindering at the same time the nuclear translocation of ERK5 [41]. Interestingly, ERK1/2-driven upregulation of DUSP1, DUSP5, and DUSP6 has been reported to inhibit ERK5 activation, suggesting the existence of a negative-feedback regulation [42]. Consistent with this idea, pharmacological inhibition of ERK1/2 in melanoma cell lines resulted in a significant ERK5 upregulation [43]. Furthermore, ERK1/2 abrogation in Colorectal cancer (CRC) has been reported to trigger ERK5 compensatory activation [44]. Notably, miR-211-mediated DUSP6 downregulation resulted in an increased ERK5 phosphorylation and BRAF inhibitors resistance in melanoma [45]. Similar results have also been documented in Non-small-cell lung carcinoma (NSCLC) cell lines, where the downregulation of DUSP6 resulted in increased ERK5 activation and epithelial–mesenchymal transition (EMT), which was reversed by inducing DUSP6 re-activation [46]. On the other hand, the Ser/Thr phosphatases PP1 and PP2A are reported to act as positive regulators of ERK5 phosphorylation [47].

In the inactive state, ERK5 is retained in the cytosol in a closed conformation, determined by the interaction between the N-terminus, the C-terminal half [48], and the chaperones Hsp90 and Cdc37, which inhibits its catalytic activity and generates a nuclear export signal [49]. Since the inhibition of Hsp90 or Cdc37 results in a rapid ERK5 ubiquitylation and proteasomal degradation, these interactions can contribute to ERK5 stability [49]. MEK5-dependent phosphorylation of the TEY motif determines the activation of the kinase domain, which in turn mediates the phosphorylation of the C-terminal half at multiple residues and the subsequent dissociation from Hsp90, exposition of the NLS, and nuclear translocation. In addition, it has been recently reported that Small Ubiquitin-related Modifier (SUMO) modification represents a necessary event for HSP90 dissociation in response to MEK5 activation and nuclear shuttling of ERK5 [50].

Once in the nucleus, ERK5 can activate either through direct target phosphorylation or through the TAD located in the C-terminal tail, a variety of genes and transcription factors such as MEF2 (A, B, and C), Activator Protein 1 (AP-1), c-Fos, c-Myc, Fos-related antigen 1 (Fra-1), Promyelocytic Leukemia Protein (PML) and Nuclear Factor kappa-light-chain-enhancer of activated B cells (NF-κB) [51,52,53,54,55,56,57]. Notably, ERK5 can phosphorylate PML and inhibits its tumor-suppressor function through activation of p21 [11]. Furthermore, it has been observed that MEK5/ERK5 hyperactivation contributes to NF-κB mediated human colon cancer (CC) progression [58] and that MEKK3 hyperactivation in ovarian cancer and glioma cells resulted in increased NF-κB activation and resistance to chemotherapeutic drugs [37,59].

Similarly to other MAPK family members, ERK5 can dock and phosphorylate its substrates by recognizing amino acids Ser or Thr followed by a Pro residue (-X-Ser/Thr-Pro-X-sequence), though Mody et al. [60] demonstrated that ERK5 can also recognize Ser/Thr sites not directly preceding Pro residues, as in the case of ERK5 C-terminus transactivation and ERK5-dependent MEK5 phosphorylation.

The phosphorylation of the C-terminus has been described as a key step for nuclear shuttling, and in particular, it has been recently observed that Thr732 acts as a functional gatekeeper residue controlling C-terminal-mediated nuclear translocation and transcriptional enhancement [61]. Furthermore, it has been reported that the C-terminus truncated construct of ERK5 (1–490 aa) is retained in the cytosolic complex with Hsp90 and Cdc37 even after ERK5 activation, since the autophosphorylation of the C-terminal tail represents a crucial event for the release of Hsp90 and the nuclear translocation of the protein [49].

Interestingly, it has been observed that the phosphorylation of the Ser730-Glu-Thr732-Pro motif could occur independently from the phosphorylation state of the TEY microdomain in the activation loop. In fact, in recent years, other mechanisms of translocation, independent from the canonical MEK5-mediated phosphorylation of the N-terminus, have been identified. During mitosis, ERK5 is phosphorylated at multiple residues within the C-terminal region [62]; these events represent a non-canonical pathway controlling ERK5 C-terminal phosphorylation. In mitosis, which involves kinase activities distinct from MEK5, Cyclin Dependent Kinase 1 (CDK1) is considered to be responsible for these phosphorylation events since its inhibition reverses mitotic phosphorylation of ERK5 [62]. ERK1/2 and CDK5 can also mediate the C-terminal domain phosphorylation in a similar manner, and subsequently induce ERK5 shuttling into the nucleus [56,63]. In particular, it has been observed that CDK5 upregulation in CRC is able to promote carcinogenesis by modulating the ERK5-AP-1 axis [56]. Besides kinase activities, the overexpression of Cdc37 can induce nuclear translocation of ERK5 independently from the phosphorylation state of the TEY motif, which could represent a mechanism exploited by many tumors; indeed, ERK5 and Cdc37 seem to cooperate to promote PC3 adenocarcinoma cell proliferation [49].

Interestingly, ERK5 mRNA expression is also subject to micro RNA-mediated post-transcriptional regulation. In particular, several reports have underlined an inverse correlation between the levels of ERK5 and the expression of miR-143. As a matter of fact, the downregulation of miR-143 usually results in a significant overexpression of ERK5, as in the case of different types of malignancies such as prostate cancer (PCa) [64,65], bladder cancer [66], BC [67,68], esophageal squamous cell carcinoma [69], nasopharyngeal carcinoma [70], acute myeloid leukemia [71], T-cell leukemia Jurkat cells [72], and B-cell malignancies [73]. In addition, it was recently demonstrated that restoring the expression of miR-200b-3p in glioma cells determined a drastic reduction of cell proliferation and EMT as a result of ERK5 suppression [74].

### 1.3. ERK5 and Cancer Proliferation

The involvement of ERK5 in tumor cell proliferation and cell cycle regulation has been widely supported by several works over the last years [75,76]. The first evidence that ERK5 was involved in regulating cell proliferation was provided through discovering that serum was a potent inducer of proto-oncogene c-JUN gene transcription via ERK5-induced MEF2C transcriptional activation [8]. Consistent with this study, it has been demonstrated that mitogens, including EGF, granulocyte colony-stimulating factor (G-CSF) and macrophages colony-stimulating factor (M-CSF), transmit their growth promoting signals via ERK5 [22,40,77]. ERK5 signaling cascade is considered an important player of cell proliferation via the expression of several mitogenic signals such as cyclin D1, SGK c-MYC, RSK2, n-MYC, and NF-κB [2,3,4,5,6,7,78]. Besides regulating the expression of proto-oncogenes required for cell growth, ERK5 is involved in sustaining the activation of survival signaling pathways [10]. Emerging data highlighted the regulative role of ERK5 in different phases of the cell cycle. In fact, by the expression of cyclin D1 and the suppression of CDK inhibitor p21, ERK5 mediates G1/S transition [3,79]. Moreover, ERK5 regulates G2/M transition and M phase by inducing the transcription factor NF-kB, which enhances the expression of mitosis-related genes, including cyclins B1 and B2 and phosphatase 2 (CDC25B) [6,62]. Recently, ERK5 has become an interesting molecule to study as a promising therapeutic target given its role in tumorigenesis and tumor malignancy [76]. The inhibition of ERK5 has been reported to decrease the proliferation rate and to intensify the number of G0/G1 cells in hepatocellular carcinoma via a mechanism involving the upregulation of p27 and p15 [9]. In addition, Tusa et al. [80] also observed that the Hedgehog (HH)-GLI signaling axis can enhance the transcription and the activation of ERK5, and that shRNA-mediated suppression of ERK5, reversed the HH-GLI-dependent proliferation of melanoma cells. In LC, characterized by an increased MEK5/ERK5 signaling axis phosphorylation, was recently demonstrated that pharmacological and genetic inhibition of both proteins reduced LC cell proliferation [39]. Shukla et al. [81] reported a link between asbestos-related ERK5 upregulation and proliferation of mesothelioma cells, as well as the inhibition of proliferative-related genes after ERK5 blockade. These findings strongly involves ERK5 as an innovative target for anti-cancer therapy. Furthermore, knockdown experiments indicated that ERK5 silencing negatively affects the proliferation of triple negative breast cancer (TNBC) cells, revealing a relevance for ERK5 in TNBC cell growth [82]. In PCa, overexpression of ERK5 enhanced the proliferation index of tumor cells, and interestingly the blockade of ERK1/2 was not sufficient to reduce the proliferation rate of tumor cells compared with ERK5 inhibition [83]. Studies on intestinal organoids demonstrated that the pharmacological inhibition of both ERK1/2 and ERK5 pathways better decreased tumor growth and the proliferation of human CRC cells [44]. Other previous evidence demonstrated that MEK5/ERK5 overexpression elicits the proliferation of CC cells and that the activation of the CDK5-ERK5 axis is required for the growth of CRC cells [56,84]. The requirement of RAS for mediating ERK5 activation downstream of growth factor stimulation remains controversial [85]. It has been reported that in a set of CRC cell lines with oncogenic KRAS-BRAF or *ERK5* amplification, ERK5 did not contribute to tumor proliferation and no evidence about the regulation of ERK5 by the mutated KRAS/BRAF signaling was observed [86]. However, recent findings showed that, in melanoma samples, mutated BRAF positively regulates activation of ERK5 that was highly correlated with melanoma proliferation by in vitro and in vivo studies [87]. Lately, Tubita et al. [88] observed a blockade in the cell cycle progression in ERK5 knockdown in BRAF-mutated melanoma cells linked to the activation of cellular senescence mechanisms which includes the involvement of CDK inhibitors. Additionally, *NRAS*-mutant melanoma cells showed a proliferative advantage when, in response to MEK inhibitors treatment, expressed high levels of active ERK5 and the high rate of ERK5 was correlated with nuclear localization of the stem-like factor KLF2 [43]. Furthermore, the presence of *MAPK7* gene amplification has been considered as a driver of proliferation in different tumor cell lines [89].

### 1.4. Role of ERK5 in Cancer Migration and Metastasis

Although most ERK5 targets are nuclear substrates, ERK5 is also a crucial regulator of cytosolic targets involved in cytoskeleton remodelling, like Akt, p90RSK, SGK, and the focal adhesion kinase (FAK) [2,90,91,92]. Interestingly, it has been demonstrated that ERK5-signaling suppression in TNBC and PC-3 cell lines lead to a drastic decrease in FAK phosphorylation, motility, and cell adhesion [93]. In accordance, recent studies have further underlined the critical linkage between the MEK5/ERK5 axis and the maintenance of the invasive capability of TNBC [94,95], LC, and melanoma [96] through FAK activation. Additionally, it has been found that ERK5 mainly localizes in the cytoplasm and membrane of ERα-negative BC cells, promoting actin remodeling, cell mobility, and invasion [97]. Moreover, suppressing the MEK5/ERK5 pathway completely prevented the TGFβ-induced EMT in murine BC cells, forcing at the same time highly metastatic tumor cells into a differentiated epithelial state [98]. In addition, ERK5 contributes to BC cell migration as an effector of the Breast tumor kinase [99] Cdc42 [100] and SRC associated in mitosis of 68 kDa signaling pathways [101]. Different reports have also highlighted the requirement for the upstream ERK5 activator MEKK2 in the migration, motility, and focal adhesion stability of invasive BC cell lines [102,103]. It is worth noting that MEK5 overexpression in BC cells can also determine a TNF-α chemotherapy-resistant phenotype characterized by the upregulation of several distinct EMT-associated genes [104]. Furthermore, STAT3 upregulation has been reported to significantly increase the transcription of MEK5, consequently enhancing BC invasiveness and metastasis formation [33]. Moreover, ERK5 has been reported to sustain the mesenchymal and migratory phenotype of TNBC cells by modulating FRA-1 expression [105] and regulating matrix-associated genes, integrins, and pro-angiogenic factors [106]. Strikingly, as revealed by in vivo studies on an orthotopic mouse model, MAPK7 silencing consistently reduced the number of circulating tumor cells and the insurgence of lung metastases [107]. Other than BC, ERK5 was also reported to be a crucial mediator of migration and invasion in osteosarcoma cell lines, regulating the expression of Slug and MMP-9 [108,109,110]. Furthermore, it has been documented that ERK5 activation is required for Src-mediated transformation and actin cytoskeleton disruption in NIH3T3 cells [30]. In addition, a series of in vitro experiments revealed that ERK5 pharmacological inhibition significantly decreased the invasiveness of HER-2-overexpressed meningioma cell lines [111]. ERK5 and its molecular target AP-1 have also been recently associated with the benzidine-induced EMT mechanism among bladder cancer cells [112]. Furthermore, ERK5 is implicated in the mechanism of podosomes formation of Src-transformed fibroblasts, inducing the Rho GTPase-activating protein 7 and thereby limiting Rho activation [113]. Moreover, pcDNA-MAPK7-transfected ovarian cancer cell lines displayed a stark increase in invasiveness and migration due to type II collagen upregulation, which was subsequently reversed through a MAPK7 silencing approach [114]. Similarly, the pharmacological inhibition of MEK5 using the specific kinase inhibitor BIX02189 resulted in a drastic reduction of cell migration in LC cells due to the suppression of the TGF-β1-induced EMT [115]. A significant association between tumor stage, presence of lymph-node metastasis and ERK5 expression was also identified by using a whole-transcriptome chip to a set of 35 primary oral squamous cell carcinomas (OSCC) [96]. In line with these observations, ERK5 expression was found to be increased in almost 60% of patients affected by clear cell renal cell carcinoma (RCC), and associated with the presence of metastasis and more advanced tumor stages [116]. Interestingly, it has been reported that triggering the EGF/ERK5/MEF2/DNA damage induced apoptosis suppressor (DDIAS) pathway enhanced cancer cells invasion through the expression of β-catenin target genes [117]. In addition ERK5 has also been identified as a downstream effector of the Protein-tyrosine kinase 6-p130 CRK-associated substrate axis, playing an important role in cancer cells migration and invasion [118]. Moreover, it has been recently demonstrated that restoring the expression of miR-200b-3p [74] and miR-429 [119] in glioma cells determined a drastic reduction of cell migration and EMT as a result of ERK5 suppression. In a similar manner, the upregulation of miR-143 in BC induced a marked reduction of ERK5 expression, proliferation, and migration among tumor cells [120]. Although a great part of the current literature describes ERK5 as a promoter of EMT and cell migration, Chen et al. [121] identified an opposite function of ERK5 in BC cell line 4T1, which could act as a metastasis suppressor by regulating the mTOR/Akt signaling.

### 1.5. ERK5 and Cancer Stemness

Compelling evidence about the role of ERK5 in the regulation of staminality has been emerging since the beginning of the last decade. By using small-molecule inhibitors and a CRISPR/Cas9 silencing approach, Williams et al. [122] demonstrated that ERK5 is involved in the activation of several pluripotency factors, including the Kruppel Like Factor 2 (Klf2), the Estrogen Related Receptor Beta (Esrrb), and Rex1. Interestingly, it was also observed that the TAD of ERK5, which is not essential for the kinase activity of the protein, is required for the maintenance of staminality through the activation of *Mef2*- and/or Sp-family transcription factors. Strikingly, recent studies have revealed a novel function of ERK5 in the process of stem cells rejuvenation. In particular, ERK5 is able to induce the transcription of the rejuvenation factor Zinc finger and SCAN domain containing 4 (ZSCAN4) at the early embryonic 2-cell stage (2C), consequently increasing telomere length and prolonging the viability of the cell culture [123]. In addition, ERK5 activity has been found strictly associated with the self-renewal of stem cells. Overexpression of Akt with angiopoietin-1 (Ang-1) determined the upregulation of ERK5, which drove transcriptional activation of cyclin D1 and Cdk4 and enhanced stem cell proliferation [124]. In line with these observations, Tusa and colleagues demonstrated that targeting the MEK5/ERK5 pathway in chronic myeloid leukemia (CML) led to a drastic decrease in progenitor/stem cells number due to a cell cycle block in G0/G1 [125]. Furthermore, the combined effect of ERK5 inhibitors with imatinib significantly reduced the expression of stemness-related genes such as c-MYC, SOX2, and NANOG [125]. Moreover, it was recently documented that the MEK5/ERK5 signaling axis is also responsible for maintaining a stem-like phenotype in CRC cells. Blocking ERK5 not only reduced the expression of pluripotency markers such as SOX2, NANOG, and OCT, but also hindered multicellular sphere formation and increased the sensitivity of CRC cells to 5-fluorouracil-based chemotherapy [126]. At last, a bivalent ERK5 inhibitor blocking the MEK5/ERK5 interaction, as well as the binding of ATP, has been developed in recent years. Strikingly, the compound was able to inhibit different cancer stem cells (CSCs) activities, such as colony formation, proliferation, and migration [127].

### 1.6. ERK5 and Cancer Cell Metabolism

The connection between the MEK5–ERK5 axis and cell metabolism has been reported [128,129,130]. In particular, it has been documented that ERK5 is essential for cell survival in oxidative phosphorylation (OXPHOS) conditions [129,131,132,133] and the MEF2 family of transcription factors mediates most of the effects of ERK5 on metabolism. Furthermore, transcriptome analysis showed that the ERK5 pathway regulates the expression of several genes involved in tumor cell metabolic remodeling [134] by controlling hypoxia-responsive genes. A recent study in pancreatic ductal adenocarcinoma identified a link between MEK5/ERK5 and the stability of MYC, a regulator of cell metabolism and growth [135]. More recently, transcriptomics analyses identified a role for the MEK5/ERK5 axis in the metabolism of Small-cell lung cancer (SCLC) cells, including lipid metabolism [136]. In-depth lipidomics analyses showed that in the absence of MEK5/ERK5, several lipid metabolism pathways are perturbed, including the mevalonate pathway that controls cholesterol synthesis. In addition, a connection between MEK5-ERK5 signaling and pyruvate metabolism and glyoxylate and dicarboxylate metabolism, as well as the citrate cycle and glycolysis, was observed. Notably, depletion of MEK5/ERK5 sensitized SCLC cells to pharmacological inhibition of the mevalonate pathway by statins [136], suggesting possible future therapeutic avenues for SCLC treatment.

The metabolic switch to OXPHOS controls the expression of ATP-binding cassette (ABC) transporters in wild-type p53-expressing cells. ABC transporters control the export of drugs from cancer cells and render tumors resistant to chemotherapy, playing an important role in multiple drug resistance (MDR). In this context, Belkahla et al. [137] found that OXPHOS increased the expression of ABC transporters in mutated (mut) or null p53-expressing cells and OXPHOS induced expression of the ERK5. Since *ABC* transporter promoters contain binding sites for the transcription factors MEF2, NRF1, and NRF2, which are targets of ERK5, the authors observed that decreasing ERK5 levels in mutp53 cells inhibited ABC expression. These results showed that the ERK5/MEF2 pathway controlled ABC expression depending on p53 status, suggesting a link between ERK5 signature, cancer cell metabolism, and MDR. Although all these data identify a new MEK5/ERK5–lipid metabolism axis that promotes cancer growth, the molecular mechanisms underlying the role of ERK5 on cell metabolism in the tumor microenvironment (TME) remain to be investigated.

## 2. ERK5 Gene Expression among Human Cancers

The level of expression and phosphorylation of ERK5 has been explored in human cancer cell lines and mouse models; on the contrary, data obtained from human tumor tissues are still limited. ERK5 is consistently expressed in several tumor types, including melanoma, osteosarcoma, mesothelioma, RCC, hepatocellular carcinoma (HCC), brain tumors, BC, LC, cholangiocarcinoma (CCA), and PCa (Table 1). Compared to their normal counterparts, neoplastic cells express higher levels of ERK5, suggesting a role for this kinase in cell transformation and tumor progression.

By gene expression profiling, *ERK5* mRNA is overexpressed in primary brain tumors, particularly in glioblastomas (GBM) [138], and in TNBC, a subgroup among primary BCs [139]. Although variable among samples, the ERK5 mRNA level is higher in LC tissue samples than in normal lung tissue [39,140]. In a small cohort of osteosarcoma patients, ERK5 overexpression has been found in most cases. Gene expression profiling in primary OSCC and oral mucosa pointed to the upregulation of MAPK signaling, among other cancer-related pathways, at this site [96].

Van Dartel et al. [141] found that most high-grade osteosarcomas had complex amplification profiles in the region 17p11.2–p12 and confirmed ERK5 amplification. Lately, a smaller common amplification region, including ERK5, has been defined at 17p11 in HCC. The copy number increase of ERK5 was confirmed in 53% of the primary HCCs [142]. ERK5 amplification has also been reported in Skin Cutaneous Melanoma (SKCM) in the TCGA database [87] and detected in 4% of the NSCLC by FISH analysis [89]. Finally, copy number gain or amplification of ERK5, corresponding to higher mRNA expression of the ERK5 gene, has been reported in 18.9% and 13.5% of lung adenocarcinomas (LUAD) and lung squamous carcinomas (LUSC), respectively [39].

### 2.1. Clinical Significance of ERK5 Gene Expression in Solid Tumors

Published in silico analysis of the TCGA datasets indicated recurrent mRNA upregulation of ERK5 in SKCM [87] and among BC subtypes, with the highest mRNA expression detected in basal-like BC [95]. The clinical relevance of ERK5 expression in terms of patient outcomes was also explored in TCGA and GEO datasets. In general, high ERK5 expression predicts poor survival outcomes. Particularly, the level of ERK5 expression inversely correlates with relapse-free survival (RFS) and distant metastasis-free survival (DMFS) in the basal-like, including TNBC and HER2^+^ BC patients [95] who received chemotherapy [139]. Furthermore, by using the Kaplan-Meier plotter tool, the mRNA expression of ERK5 has been associated with a worse prognosis in patients with node positive basal-like BC or HER2^+^ tumors [82]. In Silico analysis also revealed that ERK5 mRNA overexpression is a predictor of poor response to systemic treatments in BC. Exploring the Repository of Molecular Brain Neoplasia DaTa (REMBRANDT), including 524 glioma cases, high ERK5 mRNA expression was associated with increased tumor grade and poor overall survival (OS) [138]. In two independent datasets of CRC patients, the TCGA and GEO, high ERK5 mRNA expression correlated with worse OS [84]. Furthermore, ERK5 expression and NF-κB activation in CC resulted significantly increased in advanced tumor stages (>N2 and M1) [58].

In the osteosarcoma patients treated with chemotherapy, the ERK5 overexpression was associated with the presence of metastasis, low Huvos grade, poor treatment response, and worse OS [143]. In LC patients, high levels of ERK5 mRNA expression are associated with a worse prognosis, and this relationship is stronger among the LUAD cohort [39,140]. Specifically, the analysis of pooled data from the LUAD cohorts, through the KM-Plotter tool, found a reduced disease-free survival (DFS) with high ERK5 expression [140]. In addition, survival analysis of a cohort of 720 LUAD patients showed that combined MEK5 and ERK5 expression at high levels are significantly associated with a poor OS. Remarkably, the median survival time was reduced by almost half in the patients with high MEK5/ERK5 expression when compared to the low MEK5/ERK5 expression group [39]. Gene copy number analysis highlights that ERK5 gains or amplifications are associated with worsened clinical outcomes in LUAD, but not in LUSC. On the contrary, survival analysis failed to show any association between ERK5 gene deletion and patient outcome [39]. In melanoma, the *ERK5* upregulation and amplifications are associated with a shorter DFS and OS [87].

Interested in taking into account possible interactions with tumor types and Stage for the association with the OS, we also explored ERK5 expression in the TCGA dataset. The analyzed primary tumor types included Bladder Urothelial Carcinoma (BLCA), Breast invasive carcinoma (BRCA), Colon adenocarcinoma (COAD), Head and Neck squamous cell carcinoma (HNSC), LUAD, LUSC, Ovarian serous cystadenocarcinoma (OV), SKCM, and Uterine Corpus Endometrial Carcinoma (UCEC). Raw counts were downloaded from GDC Legacy Archive (hg19) using TCGAbiolinks R/Bioconductor package. Normalized expression levels were obtained by upper quartile normalization measured in RSEM, and the log_2_-RSEM gene expression was considered for downstream analysis (N = 3850 cases, Appendix A). The ERK5 expression was heterogeneous across tumor types (*p* < 0.0001) with the highest expression among SKCM and the lowest in BRCA samples (Figure 2A); moreover, a significant higher ERK5 expression was observed in higher stage tumors (Stage III or IV, *p* < 0.0001; Figure 2B).

In the whole cohort, the 37.1 percentile of ERK5 expression was identified as the optimal cut-off point (*p* < 0.0001, by a maximally selected rank statistic on the log-rank test) for OS analysis. The group of patients with a high ERK5 expression was associated with a significantly worse OS over time (*p* < 0.0001, Figure 2C). The evidence of ERK5 gene expression being a risk factor for the OS was also confirmed by testing its expression as a continuous variable in the univariable (*p* < 0.001, Figure 2D) Cox proportional-hazards model. The multivariable model was designed adjusting for Stage and Site effect and including interaction terms between ERK5 expression and Tumor Site. Notably, among HNSC and UCEC, ERK5 expression resulted protective for OS (Figure 2E), whereas for the other tumor sites is a clear risk factor, as shown in Figure 2E,F. Of note, among BRCA tumors, the ERK5 expression was significantly heterogeneous among different molecular subtypes, as defined by Berger et al. [144] and retrieved through TCGAbiolinks, being the Normal and Basal subtypes with the highest ERK5 expression (Figure 2G). At any rate, for this tumor site, no significant association was seen between ERK5 expression and OS (*p* = 365), as suggested by Figure 2F, nor was any significant interaction with molecular subtype (*p* = 266) seen; thus, no different association with OS was observable among different molecular subtypes (Figure 2H), which still represent a prognostic feature (*p* = 0.027) together with the overall stage (*p* < 0.0001).

### 2.2. Localization and Clinical Significance of ERK5 Protein Expression in Human Cancers

The ERK5 protein expression in human cancers has been investigated in more than twenty studies and thirteen cancer types. In LC, a strong protein expression has been reported in cases bearing ERK5 gene amplification, but also in 20% of cases without gene amplification, suggesting an alternative mechanism of ERK5 regulation [89]. In a cohort of 84 BC patients, analyzed by immunohistochemistry (IHC), the ERK5 overexpression was found in 20% of cases. In detail, the expression of ERK5 was reported as diffuse or perinuclear but limited to the cytoplasm. The authors observed ERK5 staining in the majority of the endothelial cells and stromal fibroblasts, as well. The degree of activation of ERK5 was tested using an antibody recognizing the dually phosphorylated form of ERK5 at the Thr-Glu-Tyr microdomain (pERK5) present in its activation loop as well as total ERK5 by western blotting (WB), concluding that ERK5 activation is frequently observed and detected at variable levels in the majority of the samples [145]. The pERK5 expression in BC was also observed by Antoon et al. [146] using IHC. WB showed that MEK5 and ERK5 proteins were significantly overexpressed in tumor samples in 23 LUAD [147]. IHC showed consistently higher levels of ERK5 and pERK5 in LUAD compared to normal lung tissues on tissue microarray (TMA), and pERK5 was higher in stage III–IV LC than in stage I–II [140]. Another study documented the increased cytoplasmic staining of pERK5 in transformed mesothelial cells compared to normal pleura; no differences in pERK5 expression were observed among histological subtypes [148]. A significant correlation between MET, ERK5, and the doublecortin-like kinase (DCLK1) expression was found through IHC on TMA of 73 malignant mesotheliomas; higher expression of MET, ERK5, and DCLK1 was detected in 67.1%, 48%, 50.7%, respectively compared to normal pleura showing low or absent reactivity for the three proteins [149]. ERK5 protein was strongly expressed in the cytoplasm of normal bile ducts and ductules but scant in hepatocytes; however, a significant increase in the level of ERK5 protein was observed in 25.6% of HCC [142]. A marked increase in ERK5 nuclear localization was found by Rovida et al. [9] in cirrhotic tissue, as well as in adjacent HCC, compared to normal liver.

An aberrant MEK5/ERK5 signaling might contribute to human CC progression via NF-κB activation. Accordingly, MEK5, ERK5, and NF-κB proteins overexpression was found in carcinomas arising from pre-existing adenomas [58]. A functional and mechanistic link between CDK5 and the oncogenic ERK5–AP-1 signaling pathway led to a poor clinical outcome in CRC. Through WB analysis in a small set of CRC samples, CDK5 expression was found significantly correlated with Thr732 ERK5 phosphorylation (r = 0.683 *p* = 0.042) [56].

The upregulation of ERK5 was found in PCa compared to benign prostatic hyperplasia. Particularly, in high-grade carcinomas, ERK5 immunoreactivity demonstrated both cytoplasmic and nuclear patterns, while in benign prostate tissue, ERK5 signals resulted exclusively cytoplasmic, in keeping with activation of the MEK5/ERK5 pathway in PCa [150]. Ramsay et al. [151] examined ERK5 expression by IHC on TMA, including normal prostate samples, benign prostatic hyperplasia samples (BPH), prostatic intraepithelial neoplasia (PIN), as well as primary PCa and metastatic PCa. Importantly, compared to normal prostate, nuclear ERK5 immunoreactivity was more increased in metastasis than in BPH and primary PCa. Pre-malignant PIN lesions also showed significant upregulation of ERK5 expression compared with BPH control, proposing ERK5 as an early event in prostate carcinogenesis. Finally, ERK5 overexpression was found in 58% of RCC cases [116], in gliomas [138], and TNBC [82] compared to the corresponding normal tissues. The overexpression of active phosphorylated form of ERK5 was found in one-third of OSCC TMA [96]. In primary and metastatic melanoma tissues, ERK5 protein is variably expressed [87].

ERK5 and pERK5 expression were increased in advanced-stage tumors, suggesting ERK5 as predictive biomarkers of disease progression. Particularly, high pERK5 expression was associated with advanced stage in OSCC patients [96]. ERK5 expression was also strongly associated with metastatic spread in BC’s advanced stage [95]. In two independent cohorts of patients diagnosed with RCC, a high level of ERK5 was detected in around 50% of subjects, and its expression was positively correlated with more aggressive tumor stages and the presence of distant metastasis formation [116,152]. Nuclear ERK5 expression is an independent prognostic marker in PCa, associated with worse outcomes and transition to hormone insensitive disease, whereas ERK5 cytoplasmic expression correlated with the presence of bone metastases, extracapsular disease, and high-grade carcinomas, as measured by Gleason score [83]. The prognostic value of ERK5 protein overexpression in human BC was also independently investigated by Montero et al. [145] and was associated with a significant decrease in DFS.

**Table 1 cancers-14-00348-t001:** ERK5 expression in human cancers and clinical outcome.

TUMOR TYPE	PATIENTS N°	TECHNIQUE	REAGENT/BIOINFORMATIC TOOL	OUTCOME *	REFERENCE
SKCM	479	TCGA (genomic alterations)	cBioportal	poor DFS	[87]
	66	IHC–TMA	anti-ERK5 (mMs, C-7; SCB)		
LGG and HGG	524	REMBRANDT (mRNA)		poor	[138]
	162	IHC–TMA	anti-ERK5 (mMs, C-7; SCB)	T+	
APC	1	IHC	anti-ERK5 (pRb; CST)	-	[153]
APC	11	IHC	anti-ERK5	-	[154]
	11	RT-PCR			
	11	WB	anti-ERK5 (CST)		
CCA	104	GEO (mRNA)		Portal invasion	[155]
		IHC–TMA	anti-ERK5 (polyclonal)	T+	
OSCC	35	whole-transcriptome chip			[96]
		RT-qPCR			
	306	IHC–TMA	anti-pERK5 (Thr218/Tyr220, mRb; CST)	T+; LN+	
BC	252	mRNA	R2: Genomics Analysis and Visualization Platform	poor	[156]
BC	84	IHC	anti-ERK5 (791–805 C-term, Rb; hm)	poor DFS	[145]
	23	WB	anti-ERK5 (791–805 aa C-term, Rb; hm) and anti-pERK5 (213–225 aa Erk5 C-term, Rb; hm)		
BC	39	IHC–TMA	anti-pERK5	-	[146]
OC		IHC–TMA	anti-pERK5		
BC	1803	KM Plotter		poor DMFS; M+	[95]
	514	TCGA		poor	
		IHC	anti-ERK5 (pRb; Abcam)		
BC	1809	KM Plotter (mRNA)		poor	[82]
	14	WB	anti-ERK5 (C-20)		
BC	384	Kinex™ antibody arrays (mRNA)		poor RFS and DMFS	[139]
LC (NSCLC)	74	IHC	anti-ERK5 (mRb, D23E9; CST)	-	[89]
LC	23	IHC	anti-pERK5	-	[147]
	23	WB	anti-pERK5		
LC (LUAD; LUSC)	36	IHC–TMA	anti-ERK5 and anti-pERK5 (CST)	T+	[140]
	8	RT-qPCR		poor OS	
LC (LUAD; LUSC)	1925	KM Plotter (mRNA)		poor	[39]
	972	TCGA (mRNA and CNA)	cBioportal	poor	
MESO	73	IHC–TMA	anti-ERK5 and pERK5 (pRb; CST)	-	[149]
MESO	15	IF–TMA	anti-pERK5	-	[148]
HCC	43	IHC	anti-ERK5 (polyclonal; Sigma-Aldrich)	-	[142]
	66	RT-qPCR (CNA)			
HCC	16	IHC	anti-ERK5 (polyclonal)	-	[9]
	13	WB	anti-pERK5		
CC	228	IHC	anti-ERK5 (pRb; CST)		[58]
		WB	anti-ERK5 (pRb; CST)	T+; LN+; M+	
CC	151	TCGA (RNA-Seq)	SurvExpress web resource	poor OS	[84]
	482	GEO (microarray)			
CRC	2	DNA methylation		-	[157]
CRC	9	WB	anti-ERK5 (pRb; Abcam) and anti-pERK5 (Thr732, pRb; CST)	-	[56]
RCC	50	WB	anti-ERK5 (CST)	T+; M+	[116]
RCC	19	WB	anti-ERK5 (hm or CST)	T+; M+	[152]
PCa	19	IHC	anti-ERK5 (pGt; SCB)	-	[150]
PCa	81	IHC	anti-ERK5 (pSp; hm)	poor; M+ and extracapsular disease	[83]
PCa	112	IHC–TMA	anti-ERK5	M+	[151]
OS	30	RT-qPCR		poor OS; T+; M+	[143]
OS	19	PCR (gDNA)		-	[141]
CSCC ^†^	12	IHC	anti-ERK5 SCB (pRb, C-20)	-	[158]
Mixed types ^†^	7	IHC	anti-ERK5 SCB (pRb, C-20)		
Mixed types ^†^	2	ICC	anti-ERK5 SCB (pRb, C-20)	-	[159]
	6	IHC	anti-ERK5 SCB (pRb, C-20)		

TUMOR TYPES: APC = Adrenal Pheochromocytoma; BC = Breast Cancer; CC = Colon Cancer; CCA = Cholangiocarcinoma; CRC = ColonRectal Cancer; CSCC = Cutaneous Squamous Cell Carcinoma; HCC = Hepatocellular Carcinoma; HGG = High Grade Glioma; LC = Lung Cancer; LGG = Low Grade Glioma; LUAD = Lung Adenocarcinoma; LUSC = Lung Squamous Cell Carcinoma; MESO = Mesothelioma; NSCLC = Non-Small Cell Lung Cancer; OC = Ovarian Cancer; OS = Osteosarcoma; OSCC = Oral Squamous Cell Carcinoma; PCa = Prostate Cancer; RCC = Renal Clear Cell Carcinoma; SKCM = Skin Cutaneous Melanoma; Mixed Types = include different types of cancer; **^†^** Tumor-Associated Macrophages expressing ERK5. TECHNIQUE: cBioportal = cBio Cancer Genomics Portal; CNA = copy number alterations; gDNA = genomic DNA; GEO = Gene Expression Omnibus metabase; ICC = immunocytochemistry; IF = immunofluorescence; IHC = Immunohistochemistry; KM Plotter = Kaplan-Meier plotter tool; REMBRANDT = REpository of Molecular BRAin Neoplasia DaTa; RT-qPCR = Real Time quantitative PCR; SNP = single nucleotide polymorphism; TCGA = The Cancer Genome Atlas database; TMA = Tissue Microarray; WB = Western Blot. REAGENT: CST = Cell Signaling Technology; SCB = Santa Cruz Biotechnology; hm = homemade; mMs = monoclonal Mouse; mRb = monoclonal Rabbit; pRb = polyclonal Rabbit; pGt = polyclonal Goat; pSp = polyclonal Sheep. OUTCOME: T+ = advanced stage tumor; LN+ = metastatic lymph node tumor; M+ = metastatic tumor; OS = overall survival; DFS = disease-free survival; DMFS = distant metastasis-free survival; RFS = relapse-free survival. * Outcome associated with high ERK5 gene or protein expression.

## 3. Role of ERK5 in Tumor Microenvironment

Although the role of ERK5 has been long overshadowed by other MAPK family members such as ERK1/2, p38, and the c-Jun N-terminal kinases (JNK), in recent years several authors have underlined the crucial contribution of this kinase to the development of an environmental niche able to support tumor progression. ERK5 participates in key processes for establishing the TME by avoiding immune surveillance, supporting angiogenesis, and even reprogramming immune cells toward a pro-tumoral phenotype. ERK5 is a key mediator of vasculogenesis and early angiogenesis, as demonstrated by the inability of ERK5 to form a complex vasculature and complete proper cardiac development in a mice knockout test [160]. Indeed, ERK5 deficiency led to embryonic mortality due to angiogenic failure and cardiovascular defects between day 10.5 and 11.5 [161]. In accordance, using an ERK5 conditional mutation in mice, Hayashi and collaborators [24] reported that ablation of ERK5 in adult mice induced blood vessel leakiness and hemorrhages in multiple organs leading to lethality within 2–4 weeks after the induction of Cre recombinase. Consistent with these observations, it has been demonstrated that pharmacological inhibition of ERK5, using XMD8-92, downregulates the basic FGF-mediated angiogenesis, thereby reducing cancer cells growth [11]. It is worth noting that ERK5 activation mediated by constitutive active MEK5 reduced the activity of Hypoxia-inducible factor 1-alpha (HIF1α) in endothelial cells under hypoxic conditions [162].

In addition, it has been observed that ERK5 plays a pivotal role in regulating the recruitment of inflammatory cells to promote skin carcinogenesis. Finegan and colleagues [163] have demonstrated that ERK5 depletion in keratinocytes can prevent inflammation-driven tumorigenesis by downregulating the production of C-X-C Motif Chemokine Ligand 1 and 2 (CXCL1 and CXCL2), two major neutrophil-recruiting chemokines, as well as the expression of (IL-1β), a major proinflammatory cytokine. Moreover, ERK5 ablation in malignant mesothelioma cells is accompanied by a significant reduction in tumor size as a result of the downregulation of critical angiogenesis-related (IL-8, VEGF) and proinflammatory (RANTES/CCL5, MCP-1/CCL2) cytokines [81]. Furthermore, recent studies on LC cells identified ERK5 as a key regulator of IL-6 secretion, which in turn determined the production of IL-12p70 by dendritic cells (DCs) and impaired the differentiation of Th1 cells [164].

ERK5 was also found implicated in the process of platelet aggregation as a central mediator of the TLR4/integrin GPIIb/IIIa pathway. This observation acquires particular importance in the context of surgical removal of solid tumors, where targeting the TLR4/ERK5 signaling axis demonstrated potential for preventing neutrophil extracellular traps (NET)-driven distant metastasis [165].

In addition, ERK5 is usually overexpressed in CD163^+^ tumor-associated macrophages (TAMs) derived from different types of malignancies such as bladder, lung, and breast cancer. In line with these observations, blocking the ERK5 pathway in TAMs promotes a phenotypic switch from a pro-tumor to an anti-tumor phenotype due to STAT3 downregulation [158]. In addition, our group demonstrated that ERK5 maintained the capacity of macrophages to proliferate by suppressing p21 expression, and TAMs proliferation is a significant feature of advanced stages of cancer [159]. Furthermore, both genetic and pharmacological inhibition of ERK5 were able to downregulate the expression of c-Myc in IL-4 stimulated macrophages, a critical transcription factor required for M2 polarization [57].

Interestingly, it has been recently discovered that cancer-associated fibroblasts (CAFs) can induce the activation of ERK5 in CRC cells. ERK5 upregulation stimulates the expression of PD-L1 in the tumor, thereby inhibiting apoptosis and promoting cancer progression [166]. Moreover, it has been observed that shRNA-mediated ERK5 depletion reduced the expression of Major histocompatibility complex (MHC) class I molecules in mouse leukemia cells, thus allowing their detection and elimination by activated natural killer (NK) cells [167]. Similar results have also been reported in breast and colon cancer. Restoring the expression of miR17/20a reduces the production of MHC-I molecules as a result of ERK5 pathway suppression, triggering NK cell anti-tumor activity at the same time [168].

ERK5 has also been implicated in the recruitment of T-lymphocytes at the tumor site. Doubly mutant PCa tissues for the Phosphatase and tensin homolog (Pten) and ERK5 exhibited a significant upregulation of lymphocyte-recruiting chemokines (Ccl5 and Cxcl10), as well as an increase in CD4^+^ T-cell infiltrate in the tumor stroma upon ERK5 ablation [169].

Taken together, these findings all underline the importance of ERK5 in the regulation of the immune cells and the TME. Interestingly, as demonstrated by Lin and colleagues [170], the inhibition of the catalytic activity of ERK5 alone had no effect on cellular proliferation and immune response compared to a total genetic ERK5 suppression, underlying the importance of other non-catalytic functions of ERK5 in the regulation of inflammatory and proliferative networks.

## 4. Alternatively Spliced Variants of ERK5

For eukaryotic genes, the primary transcripts (pre-mRNAs) generated by RNA Polymerase II need to be extensively processed in order to remove introns and join exons to generate the protein coding of mature RNA (mRNA). This pre-mRNA splicing process, operated by a very complex RNA-protein macromolecular machinery (the spliceosome), guarantees that intron removal occurs with extraordinary precision and efficiency in the nucleus of eukaryotic cells, thus allowing only correctly processed mRNAs to translocate in the cytoplasm. Despite being a very expensive process (both in terms of energy and time), splicing provides an exquisite opportunity to fine-tune the regulation of gene expression. Indeed, the high flexibility characterizing the splicing reaction allows for 5′ or 3′ splice sites at the exon-intron junctions to be combined in different manners to produce multiple mRNA isoforms from a single pre-mRNA through a process known as alternative splicing (AS). Five main types of AS events generate mRNA isoforms with different stability, or can encode for proteins with diverse structure, function, or cellular localization (Figure 3A,B). More than 90% of human multiexon genes undergo AS regulation [171,172], whereas AS is far less frequent in lower eukaryotes [173,174]. Thus, AS represents one of the major mechanisms underlying the human transcriptome and proteome diversification, starting from a limited number of coding genes. Highly coordinated AS events sustain the generation of protein isoforms with cell- or tissue-specific expression [172]. Notably, these variants play key roles in all major aspects of eukaryotic cell biology in highly specialized tissues and organs in response to environmental stimuli [175]. In addition, AS is common in genes working in similar biological pathways, and AS regulated exons frequently encode for domains involved in protein-protein interactions. Based on this, it has been suggested that AS “switches” have important functional implications in controlling gene regulatory networks and signaling cascades [176,177].

Importantly, a large number of AS events are cancer-specific, leading to the production of protein isoforms directly involved in tumor establishment, progression, and resistance to therapeutic treatments [178,179]. In fact, these variants allow neoplastic cells to sustain their high proliferation rate, stimulate migration and invasion of the surrounding tissues, resistance to apoptosis, formation of new blood vessels, modification of metabolism to cope with a hypoxic microenvironment, and acquisition of mechanisms of immune escape. In some cases, cancer-specific AS variants are generated through mutations within the splice sites, whereas they are frequently the consequence of altered expression or activity of specific AS regulators, with some of them acting as oncogenes or tumor suppressors [178,179]. Thus, splicing alterations are to be considered as important genetic modifiers of tumorigenesis and hallmarks of cancers [179]. Notably, cancer-restricted AS changes could be particularly useful for cancer diagnosis and prognosis, whereas their targeting could constitute a promising approach to developing more selective therapies [180].

Whole-genome sequencing showed that a large fraction of the human genome encodes for protein kinases (518 genes, nearly 2% of all human genes) [181], strongly supporting the notion that phosphorylation plays a key role in regulating protein function. Intriguingly, bioinformatic analyses also revealed that the 518 protein kinases genes produce more than 900 protein kinase isoforms through AS regulation [182]. Several signaling pathway components act as oncoproteins or have a known role in human cancers [183]. Notably, AS variants of oncogenic kinases (frequently not expressed in non-pathological tissues) have been described themselves to be involved in tumor development and maintenance [184]. Thus, AS regulation of kinase-encoding genes adds an additional layer of complexity to tumor cell biology, which could be a target for therapeutic purposes.

To date, two different MEK5 isoforms (MEK5α and MEK5β) have been characterized [185]. While MEK5α was reported to promote ERK5 kinase activity, the catalytically inactive MEK5β acts as a dominant-negative regulator, preventing MEK5α-ERK5 association. As a matter of fact, it was observed that MEK5β is usually downregulated in different tumors, and that ectopic expression of this isoform arrests BC cell growth [186].

By using different bioinformatic tools (https://imm.medicina.ulisboa.pt/group/exonmine/; http://srv00.recas.ba.infn.it/ASPicDB/; http://www.ensembl.org/index.html; https://genome.ucsc.edu/, accessed on 7 January 2022), multiple ERK5 mRNAs that differ in the 5′ UTR region have been identified (Figure 3C). Moreover, alternatively spliced ERK5 transcripts with diverse coding sequences have also been recognized. Specifically, different human ERK5 mRNAs are generated through the alternative selection of different donors and acceptors splice sites within the first exons. Similar findings have also been reported by Yan and colleagues for the mouse gene (Figure 3D), which shares 91% identity with human ERK5 [19]. The consequence of these splicing decisions is the introduction of a premature stop codon promoting the usage of a downstream ATG and the production of shorter N-termini ERK5 variants (called mERK5b and mERK5c). Compared to the full-length protein, the mERK5b and mERK5c isoforms lack 69 and 139 amino acids at their N terminus, respectively. Since these deletions eliminate the cytoplasmic targeting domain, mERK5 splicing variants are exclusively localized in the nucleus differently from the full-length ERK5 present both in the cytoplasm and the nucleus. mERK5b and mERK5c, in addition to having different cellular localizations, also play different roles. Indeed, mERK5b and mERK5c do not bind ATP and are not active kinases. Importantly, they are not able to associate with MEK5 and act as a dominant-negative inhibitor of ERK5 kinase activity and ERK5-mediated MEF2C transactivation.

Analysis of the EST database also revealed the existence of another alternatively spliced murine ERK5 transcript [187]. This mRNA molecule, called Erk5-T (Erk5-Truncated), is expressed in different mouse tissues and generated from the retention of intron 4 [187]. This splicing outcome is the introduction of a premature stop codon removing from the corresponding protein the nuclear localization signal and the proline-rich domain. Similar to the full-length protein, mERK5-T is phosphorylated by MEK5. However, mERK5-T is mainly located in the cytoplasm, and it does not translocate to the nucleus upon activation. In addition, it is able to impair, in a concentration-dependent way, the nuclear translocation of activated ERK5 [187].

Taken together, these observations indicate that the expression of alternatively spliced ERK5 isoforms could provide a mechanism to control ERK5 signaling in a tissue- or developmental-stage specific manner with relevance to pathological conditions. In line with this, compared to other adult tissues, the mErk5-T transcript has also been found in bone marrow [187], thus suggesting that in a particular physiological context (for example, during hematopoiesis, in which the activity of ERK5 has been well documented [40]), mERK5-T could reduce the ERK5 nuclear functions and/or keep it in the cytoplasm for more extensive (or longer) association with its downstream targets.

A human alternatively spliced transcript similar to the mouse mERK5-T and generated from splicing retention of intron 4 also exists in the human EST database [187].

Lastly, a renal-specific variant of ERK5 with a molecular mass of 80 kDa was identified in the kidney. Although different from all previously characterized isoforms, the functional role and the structure of the ERK5 renal variant remains unknown since RT-PCR failed to detect novel ERK5 transcripts in the kidney [188].

Collectively, these findings emphasize once again that anti-cancer strategies should take into account the expression of tumor-associated alternatively spliced proteins as well as their functional role in specific biological contexts.

## 5. Conclusions

Compared with conventional MAPK family members, such as MEK1/2, the therapeutic potential of ERK5 has been largely underappreciated. Nevertheless, biochemical and functional studies in recent years have elucidated the unique structural and functional properties of the MEK/ERK5 pathway, as well as its strong association with oncogenesis and cancer progression. Over the years, several knockout studies in mice, as well as studies with human samples, have shed light on the MEK5/ERK5 axis as a novel target in cancer metastasis and therapy resistance. Here, we reported that upregulation and overexpression of ERK5 in several tumor types are associated with advanced stage, metastases, and worse overall survival, highlighting the clinical significance and the prognostic value of ERK5 deregulation in human cancer. We also presented emerging insights regarding the relevance of upregulated ERK5 in TME and its role in maintaining cancer stem-like phenotype. Identifying ERK5 as a potential anti-cancer drug target induced academic groups and pharmaceutical companies to develop small-molecule ERK5 kinase inhibitors. However, it has been recently reported that selective ERK5 kinase inhibitors fail to recapitulate ERK5 genetic deletion phenotypes, suggesting kinase-dispensable functions for ERK5 [189]. On that basis, it is becoming clear that both the ERK5 kinase and TAD must be considered when assessing the role of ERK5 and the effectiveness of anti-ERK5 therapeutics. In addition, since naturally occurring ERK5 splice variants that lack either the C-terminal TAD or the N-terminus exist, it will be important to determine their expression and role in cancer cells and tumor-infiltrating immune/stroma cells. Overall, our work, as well as others [15,76,190], highlights the importance of deeply understanding the MEK5-ERK5 pathway and its variants to expand the current spectrum of target therapies in cancer treatments.

## Figures and Tables

**Figure 1 cancers-14-00348-f001:**
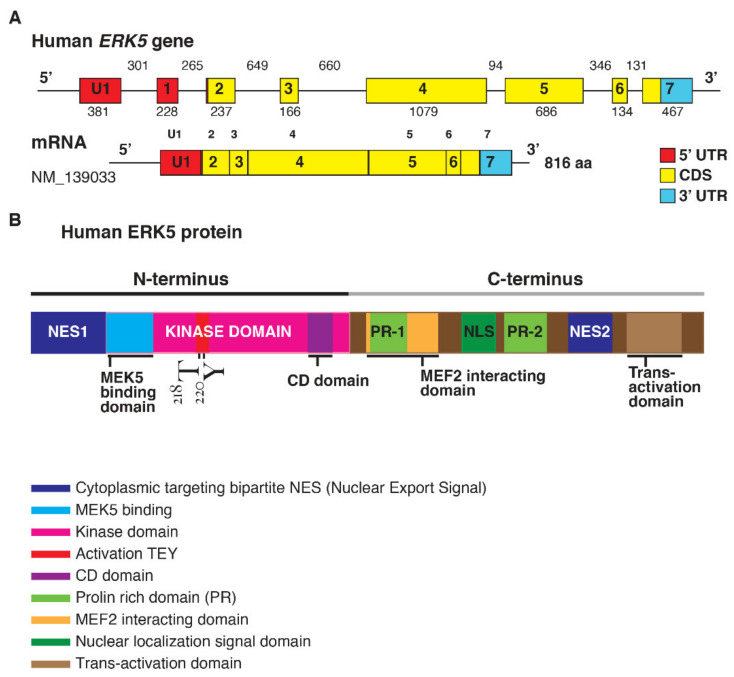
(**A**). Structure of the human *ERK5* gene (*MAPK7*) and the mRNA transcript (NM_139033) encoding for the full-length ERK5 protein of 816 aa; length of the exons (below) and introns (upper) are also shown (size not to scale). The exon-intron scheme is depicted with 5′UTR in red, the coding region (CDS) in yellow, and 3′UTR in light blue. (**B**). Human ERK5 protein and its functional domains: at the N-terminal there is a cytoplasmic targeting signal and a kinase domain, which comprises the region of binding for MEK5 and the common docking (CD) domain. Residues Thr218/Tyr220 are underlined in red as the site of MEK5 phosphorylation. The C-terminal comprises two proline-rich (PR) domains, a MEF2-interacting region, the nuclear localization signal (NLS) domain, and a transcriptional activation domain. ERK5 contains a bipartite nuclear exportation signal (NES1 and NES2). The N- and C-terminal halves of ERK5 interact, producing a nuclear export signal that retains ERK5 in the cytoplasm of resting cells.

**Figure 2 cancers-14-00348-f002:**
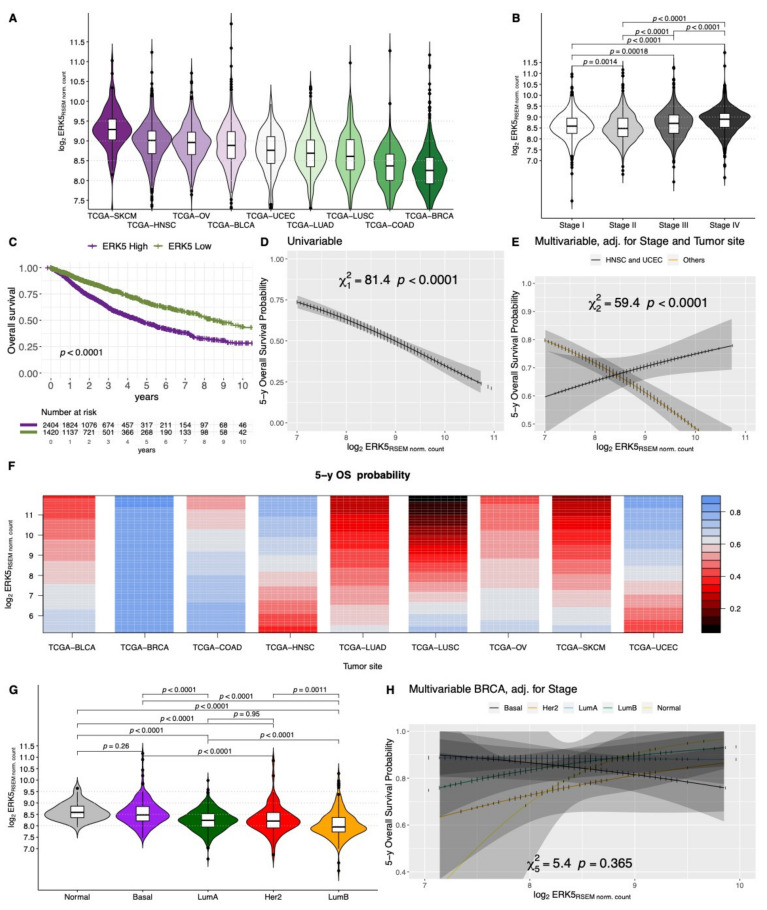
Violin plots showing the different log2 gene expression of ERK5 across different Projects of TCGA (**A**) and Stage groups (**B**), adjusted *p* values estimated by Wilcoxon test. Kaplan-Meier overall survival curves showing the comparison of the group of patients with an ERK5 gene expression above the optimum cut-off of 37.1 percentile (ERK5 High) or below it (ERK5 Low) (**C**); *p*-value estimated by adjusted log-rank test. Partial effect plots of the ERK5 expression against the estimated 5-year overall survival probability considering the univariable (**D**) or multivariable Cox model adjusted for Stage and Site including interaction with tumor site groups (HNSC or UCEC vs. Others) (**E**); *p* values estimated by Wald statistics. Contour plot representing the partial effects of ERK5 expression and projects for the estimated 5-year overall survival probability (color gradient), adjusted for Stage, including in the multivariable model a possible interaction between ERK5 expression and tumor site, meaning a possible different effect of ERK5 expression among different tumor types (**F**). Violin plots showing the different log2 gene expression of ERK5 across different molecular subtypes within the BRCA-TCGA dataset (**G**), adjusted *p* values estimated by Wilcoxon test. Partial effect plots of the *ERK5* expression against the estimated 5-year overall survival probability in the BRCA-TCGA dataset, showing the estimated effect separately in each molecular subtype (**H**). Legend: OS, Overall Survival; BLCA, Bladder Urothelial Carcinoma; BRCA, Breast invasive carcinoma; COAD, Colon adenocarcinoma; HNSC, Head and Neck squamous cell carcinoma; LUAD, Lung adenocarcinoma; LUSC, Lung squamous cell carcinoma; OV, Ovarian serous cystadenocarcinoma; SKCM, Skin Cutaneous Melanoma; UCEC, Uterine Corpus Endometrial Carcinoma.

**Figure 3 cancers-14-00348-f003:**
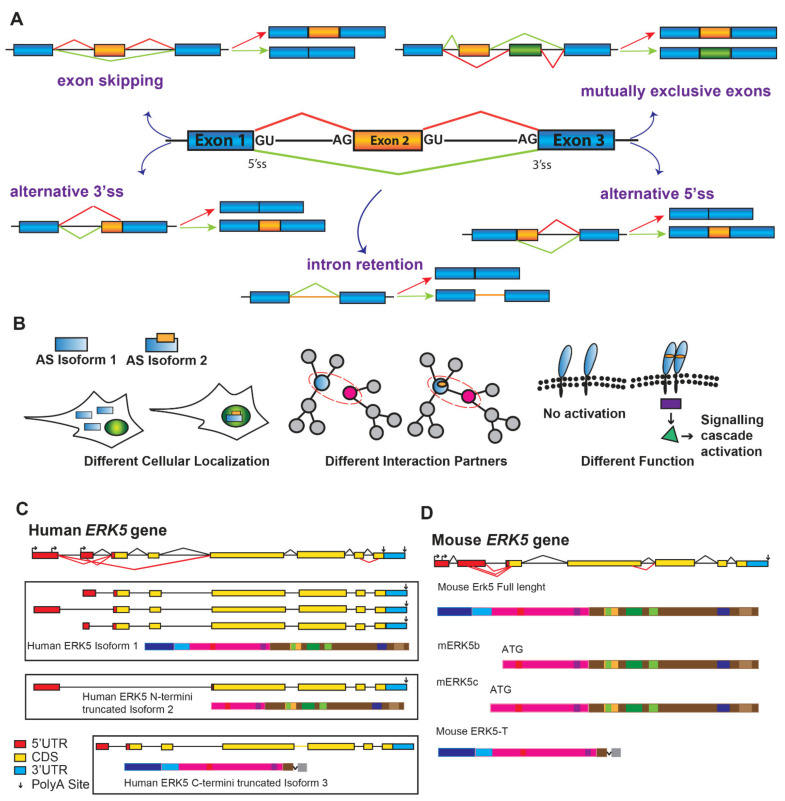
ERK5 alternative splicing variants. (**A**). Schematic representation of five possible alternative splicing (AS) mechanisms. Constitutive exons = blue, alternative exons = orange, introns = lines. Different AS mRNAs result from exon skipping, intron retention, usage of alternative 3′or 5′splice sites (3′ ss and 5′ ss), and from the selection of mutually exclusive exons. (**B**). Protein isoforms generated through AS reaction can differ for their cellular localization (cytoplasmic vs. nuclear in the case of isoform 1 and 2) (left), for their protein-protein interactions (center), or their function as components of a signaling pathway (right). (**C**). Predicted transcripts for human ERK5 gene based on supported EST adapted from ASPicDB (http://srv00.recas.ba.infn.it/ASPicDB/, accessed on 7 January 2022). The exon-intron scheme is depicted with 5′UTR in red, the coding region (CDS) in yellow, and 3′UTR in light blue; polyadenylated transcripts are marked with vertical arrows; alternative transcription start sites are also represented. The canonical ERK5 human protein, generated by three different transcripts that differ in their 5′UTR (exons in red), is shown as isoform 1. Another mRNA transcript generated by skipping of first exons can encode for a protein isoform deleted of the N-termini region (isoform 2), whereas the ERK5-C termini truncated protein, generated by intron retention, lacks the cytoplasmic targeting and MEK5 binding domains (isoform 3). (**D**). Schematic representation of murine *ERK5* gene with 5′UTR in red, CDS in yellow, and 3′UTR in light blue. Three different ERK5 murine proteins are depicted: mERK5b and c differ for their ATG and lack the cytoplasmic targeting domain (blue), mouse ERK5-T lacks the NLS and the PR2 domain.

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
