# Peer review of "Clinical Significance and Regulation of ERK5 Expression and Function in Cancer"

_cancers, 2022, doi:10.3390/cancers14020348_

Round 1

Reviewer 1 Report

In the last ten years there has been an overwhelming explosion of scientific articles that make the MEK5/ERK5 pathway visible as a very important pathway to be taken into account when designing new therapeutic strategies in cancer. Although reviews of this signalling pathway have recently been published, the present review by Matilde Monti et
al focuses on the role of deregulated ERK5 expression and activation in cancer and its importance in patient prognosis. They complete their work on the involvement of deregulated ERK5 in the tumour microenvironment, and further highlight the relevance of ERK5 isoforms in tumour growth.
This review undoubtedly contributes to the progress of ERK5 research to the different research groups of the scientific community working in the field of cancer.
However, before publication, authors should reread their work and pay close attention to:
    - check all references throughout the manuscript.
For example,
in section 1.3 ERK5 and cancer stemness, check references 83 and 84.
In Section 2. ERK5 gene expression among human cancers:
    BC: it is supposed to refer to Breast Cancer, but it would be more correct to specify it.
    reference 84 refers to Human Chronic Myeloid Leukemia Stem Cells and not to SKCM
In particular, references in table 1 are out of place, probably due to a mistake made when editing the bibliography.
    For example, the first bibliographical reference in Table 1, number 93, does not refer to the SKCM tumour type or to that number of patients, but to NSCLC and squamous esophageal cancers.
    The same applies to the second bibliographic citation. Reference 86 does not deal with LGG or HGG
    in relation to APC in table 1, it seems that the bibliographic references have moved one place.
    Please check all references in Table 1. Most, if not all, are misplaced.
Check also, in table 1, if the number of samples analysed matches the correct bibliographic reference, and the technique used. For example, on CC, were 228 patient samples analysed by WB? On BC, were 3625 samples analysed by KM Plotter?
In section 3. Role of ERK5 in Tumor microenvironment (TME)
    Please check that quote 24 refers to what you mention in the text “Cre-induced depletion of ERK5 is able to promote the regression of tumor vasculature, followed by a consequent reduction in tumor size [24].”

Author Response

Reviewer 1

In the last ten years there has been an overwhelming explosion of scientific articles that make the MEK5/ERK5 pathway visible as a very important pathway to be taken into account when designing new therapeutic strategies in cancer. Although reviews of this signalling pathway have recently been published, the present review by Matilde Monti et
al focuses on the role of deregulated ERK5 expression and activation in cancer and its importance in patient prognosis. They complete their work on the involvement of deregulated ERK5 in the tumour microenvironment, and further highlight the relevance of ERK5 isoforms in tumour growth.
This review undoubtedly contributes to the progress of ERK5 research to the different research groups of the scientific community working in the field of cancer.
However, before publication, authors should reread their work and pay close attention to:
    - check all references throughout the manuscript.
For example,
in section 1.3 ERK5 and cancer stemness, check references 83 and 84.

Response. The indicated references in the section “1.5 ERK5 and cancer stemness” have been checked and corrected (Refs 125 and 126).

In Section 2. ERK5 gene expression among human cancers:
    BC: it is supposed to refer to Breast Cancer, but it would be more correct to specify it.

Response. In section 1.2, it is already reported that BC refers to Breast Cancer. Accordingly to the 2nd Reviewer’s suggestion, we specified the abbreviations the first time we reported the name in the text, then the abbreviation was used.

reference 84 refers to Human Chronic Myeloid Leukemia Stem Cells and not to SKCM

Response. The right reference (Tusa et al. 2018 PMID: 29483645, Ref 87) has been cited in the revised version.

In particular, references in table 1 are out of place, probably due to a mistake made when editing the bibliography.
    For example, the first bibliographical reference in Table 1, number 93, does not refer to the SKCM tumour type or to that number of patients, but to NSCLC and squamous esophageal cancers.    The same applies to the second bibliographic citation. Reference 86 does not deal with LGG or HGG
    in relation to APC in table 1, it seems that the bibliographic references have moved one place.
    Please check all references in Table 1. Most, if not all, are misplaced

Response. Thank you for the corrections. Now, the right references have been reported in Table 1.

Check also, in table 1, if the number of samples analysed matches the correct bibliographic reference, and the technique used. For example, on CC, were 228 patient samples analysed by WB?

Response. 323 colon samples, including 53 normal colon, 42 adenomas, and 228 carcinomas, were included in the study by Simões et al., 2015. Representative immunoblots and immunohistochemistry were shown in the figures; no details about how many patients were analyzed specifically in WB and/or IHC were reported by the authors. We adjust the Table 1 accordingly.

On BC, were 3625 samples analysed by KM Plotter?

Response. 1803 samples instead of 3625 were assessed by KM plotter in the study by Xu et al. We corrected the Table 1 accordingly. We also revised the number of patients analyzed through different technique in each of the studies reported in Table 1. Ortiz-Ruiz et al. assessed the relationship between ERK5 expression and patient outcome, using a publicly available database including microarray expression data from 1809 breast cancer patients reported in the study by Gyorffy et al., (PMID: 20020197). No more mistakes are left.

In section 3. Role of ERK5 in Tumor microenvironment (TME)
    Please check that quote 24 refers to what you mention in the text “Cre-induced depletion of ERK5 is able to promote the regression of tumor vasculature, followed by a consequent reduction in tumor size [24].”

Response. As requested by the reviewer, we checked the Ref 24. The indicated sentence was removed and replaced with the follow one: “In agreement, using a ERK5 conditional mutation in mice, Hayashi and collaborators [24] reported that ablation of ERK5 in adult mice induced blood vessels leakiness and hemorrhages in multiple organs leading to lethality within 2–4 weeks after the induction of Cre recombinase”.

Reviewer 2 Report

"reading frame of MAPK7 is 2541bp. This is translated into a 816aa protein."

Genetic code is organized in triples. 2541/3=847aa is not 816aa. Please write the length of the mRNA, the length of the ORF on the mRNA and the length of the gene on the DNA with the length of the introns and exons.

"(PI3K/Akt) pathway is able to trigger the activation of MEK5-ERK5 in

neuroblastoma (NB) and malignant mesothelioma (MM)"

These cancer abbreviations only appear once in this article. Better not to enter these abbreviations. Especially that MM is also "Multiple Myeloma".

"the transcription of MEK5, consequently enhancing breast cancer (BC) invasiveness and"
"[72], breast cancer [73,74], esophageal squamous cell carcinoma [75], nasopharyngeal car-"
inaccurate use of abbreviations

"PC" abbreviation was not introduced into the text

"8, VEGF) and proinflammatory (RANTES/CCL5, MCP-1) cytokines [119]. Furthermore,"
the current name of MCP-1 is CCL2

The authors have written subsections about the role of ERK5 on CSC, angiogenesis and tumor associated cells. But authors need to write separate subsections on the importance of ERK5 on cancer cells proliferation, cancer associated-inflammation, cancer cells migration, cancer cell metabolism and cancer metastasis.

The article contains a small number of figures and tables.

Author Response

Reviewer 2

"reading frame of MAPK7 is 2541bp. This is translated into a 816aa protein."

Genetic code is organized in triples. 2541/3=847aa is not 816aa. Please write the length of the mRNA, the length of the ORF on the mRNA and the length of the gene on the DNA with the length of the introns and exons.

 "(PI3K/Akt) pathway is able to trigger the activation of MEK5-ERK5 in

neuroblastoma (NB) and malignant mesothelioma (MM)"

These cancer abbreviations only appear once in this article. Better not to enter these abbreviations. Especially that MM is also "Multiple Myeloma".

Response. We removed NB and MM abbreviations.

 "the transcription of MEK5, consequently enhancing breast cancer (BC) invasiveness and"
"[72], breast cancer [73,74], esophageal squamous cell carcinoma [75], nasopharyngeal car-"
inaccurate use of abbreviations

Response. As suggested, we specified the abbreviations the first time we reported the cancer type in the text, then we used the abbreviations. We did not enter abbreviations for cancer types that appeared once time.

"PC" abbreviation was not introduced into the text 

Response. “PCa” abbreviation was used to refer to prostate cancer; we did not enter “PC” abbreviation that usually refered to pancreatic cancer.

"8, VEGF) and proinflammatory (RANTES/CCL5, MCP-1) cytokines [119]. Furthermore,"
the current name of MCP-1 is CCL2

Response. We added the name CCL2.

The authors have written subsections about the role of ERK5 on CSC, angiogenesis and tumor associated cells. But authors need to write separate subsections on the importance of ERK5 on cancer cells proliferation, cancer associated-inflammation, cancer cells migration, cancer cell metabolism and cancer metastasis.

Response. As suggested by the reviewer, separate subsections including “1.3. ERK5 and cancer proliferation”, and “1.4. Role of ERK5 in cancer migration and metastasis” and “1.6. ERK5 and cancer cell metabolism” were added in the revised version. The link of ERK5 with cancer-associated inflammation is described in the “3. Role of ERK5 in Tumor microenvironment” section.

The article contains a small number of figures and tables.

Response. We added one figure (Figure 1) and two graphs in figure 2 (G, H).

Round 2

Reviewer 2 Report

The revised manuscript was much improved.

Author Response

Thank you for your comments and suggestions.